# Detection of Circulating Tumor Cells and EGFR Mutation in Pulmonary Vein and Arterial Blood of Lung Cancer Patients Using a Newly Developed Immunocytology-Based Platform

**DOI:** 10.3390/diagnostics14182064

**Published:** 2024-09-18

**Authors:** Hitoshi Dejima, Hayao Nakanishi, Ryo Takeyama, Tomoki Nishida, Yoshikane Yamauchi, Yuichi Saito, Yukinori Sakao

**Affiliations:** 1Department of Surgery, Teikyo University School of Medicine, Itabashi-ku, Tokyo 1738605, Japan; dejima14@med.teikyo-u.ac.jp (H.D.); takeyama.ryou.dm@teikyo-u.ac.jp (R.T.); nishida.tomoki.lb@teikyo-u.ac.jp (T.N.); yoshikaney@med.teikyo-u.ac.jp (Y.Y.); yuichi.saito@med.teikyo-u.ac.jp (Y.S.); ysakao@med.teikyo-u.ac.jp (Y.S.); 2Department of General Thoracic Surgery, Shin-Kuki General Hospital, Kuki 3468530, Japan; 3Laboratory of Clinical Pathology, Okazaki City Hospital, Okazaki 4440002, Japan

**Keywords:** circulating tumor cells (CTCs), cytology-based CTC detection platform, EGFR mutation analysis, digital PCR, lung cancer

## Abstract

Background: Epidermal growth factor receptor (EGFR) tyrosine kinase inhibitors are powerful molecular targeted therapeutic agents for lung cancer. We recently developed an original immunocytology and glass slide-based circulating tumor cell (CTC) detection platform for both CTC enumeration and EGFR mutation analysis with DNA extracted from CTCs. Methods: Using this platform, we conducted a pilot clinical study for CTC enumeration in peripheral blood (PB), pulmonary arterial blood (PA), and pulmonary venous blood (PV) from 33 patients with lung cancer (Stage I–III) who underwent surgery, followed by digital PCR-based EGFR mutation analysis of CTCs in PV from 12 patients. Results: The results showed that CTC levels were significantly higher in PV and PA than in PB (*p* < 0.05, *p* < 0.01. respectively), with a notably greater number of small and large CTC clusters (*p* < 0.01). Genetic analysis of EGFR mutations of CTCs from PV (*n* = 12) revealed six mutations, including three Exon19del and three L856R, in CTCs and eight EGFR mutations, including five Exon19del and three L856R, in lung tumor tissue. CTC mutation status matched that of tissue samples in nine patients, was unmatched in two patients, and controversial in one patient, indicating a sensitivity of 0.75 (6/8) and specificity of 1.0 (4/4) with some false-negative results for the mutation analysis of CTCs. Conclusions: This immunocytology-based CTC detection platform is a convenient method for detecting both CTC number and EGFR mutation status under microscopy, suggesting its potential as a liquid biopsy tool in the hospital for patients with lung cancer in some clinical settings.

## 1. Introduction

In lung cancer, a genetic test for epidermal growth factor receptor (EGFR) mutations is a critical diagnostic tool, because the clinical efficacy of first- and second-generation EGFR tyrosine kinase inhibitors (EGFR-TKI) has been well established for drug-sensitive and -resistant EGFR-mutated non-small-cell lung carcinomas [1,2]. A major problem of this genetic test is the difficulty of repetitively obtaining a sufficient amount of DNA from lung tumor tissue non-invasively, depending on tumor size and the location of the tumor in the lung [3]. In this respect, circulating tumor cells (CTCs) in blood have attracted significant attention as an alternative source of tumor cells derived from primary lung tumors [4]. Cell-free, circulating tumor DNA (ctDNA), exosome, and CTCs from patient blood are well-recognized components of liquid biopsy [5,6,7]. Compared with CTCs, ctDNA is now clinically available and widely used as a genetic test, despite challenges such as its lower sensitivity in early-stage lung cancer [8,9,10]. In contrast, CTC-derived mutation tests remain primarily in the pre-clinical stage, largely due to the rarity of CTCs in peripheral blood (PB) [11,12].

Concerning the detection of CTCs, numerous technologies have been developed. Most CTC detection platforms depend on multicolor immunofluorescence staining such as Keratin+/EpCAM+/CD45−/DAPI+ [11,12]. In contrast, reports on the immunocytology-based CTC detection method have been quite limited [13]. Recently, we developed a unique and convenient platform based on immunocytological detection of CTCs on a glass slide under light microscopy [14,15,16]. In this study, we developed a new DNA extraction method after cytokeratin immunostaining of CTCs on a glass slide through an alkaline phosphatase (AP) and LPR-based color development system. This is because AP does not induce DNA damage unlike the hydrogen peroxide (H_2_O_2_)/horseradish peroxidase (HRP) system [17,18]. Using this updated CTC detection method, we conducted a pilot study to detect EGFR mutations in lung cancer using digital PCR in 12 patients. In addition, we compared the characteristics of CTCs in PB, pulmonary arterial blood (PA), and pulmonary venous blood (PV) from 33 patients with stage I–III lung cancer who underwent surgery. We previously reported the presence of abundant CTCs in the venous blood drainage of patients with breast and colon cancer [15,16], as well as the presence of megakaryocytes in the arterial and venous blood of patients with lung cancer [19,20]. The hemodynamic aspect of CTCs in the pulmonary circulation, including PA, PV, and PB, as well as the potential diagnostic pitfalls of CTCs by megakaryocytes in PA and PB, are discussed.

## 2. Materials and Methods

### 2.1. Reagents

A mouse monoclonal antibody against human broad-spectrum (pan)-cytokeratin (Clone, Oscar) was purchased from BioLegend (Dedham, MA, USA). Immunostaining and subsequent color development system (DAKO) were performed as described later. Meyer’s hematoxylin was used for nuclear counterstaining of the cells.

### 2.2. Patients and Bloods

Patients with stage I–III primary lung cancer (*n* = 33) who underwent surgery at Teikyo University Hospital during 2021–2023 were enrolled in this study. The average age of the patients was 72 years, and the male/female ratio was 18/15. PB was collected from a cubital vein and PV/PA was obtained from the main trunk of the pulmonary vein and artery of the resected lung immediately (within several minutes) after resection. The punctured blood samples were pooled in specialized tubes for liquid biopsy (Streck, La Vista, NE, USA) and kept at room temperature. CTCs were collected by our detection platform within 24 h after puncture. The tumors ranged from stage I to III, with histology mostly being adenocarcinoma based on UICC (Union for International Cancer Control) criteria (Table 1). This study was approved by the institutional ethics review board of Teikyo University Hospital (Approval number: No.21-232), and written informed consent was obtained from each patient prior to sample collection. This study met the standards defined by the principles outlined in the Declaration of Helsinki.

### 2.3. Filtration-Based CTC Collection and Subsequent CTC Transfer to Slide Glass

This process consisted of 3 steps, as follows: (1) Filtration-based CTC collection system. This was composed of a microfluidic chip (filter chip) containing a nickel filter with 8 μm pores and a 3-dimensional (3D) lattice structure of 10 nm thickness (Optnics Precision Co., Ltd., Tochigi, Japan) (Figure 1a). Four filter chips could be run simultaneously by the automated CTC enrichment apparatus with a fluid pressure control system [15] (Maruyasu Industries Co., Ltd., Okazaki, Japan). After applying whole blood to this apparatus, CTCs collected on the filter were washed with PBS/EDTA solution and subsequently underwent fixation with 10% formalin solution for 20 min. (2) CTC transfer system to the glass slides. The 3D metal filter was detached from the filter chips and was placed upside down on a coated glass slide (MAS coat, Matsunami, Osaka, Japan) [17] and transferred to a coated glass slide by the air pressure-mediated transfer apparatus (Maruyasu Industry Co., Ltd.). (3) Immunocytochemistry of CTCs on a glass slide. CTC glass slides stored in 95% ethanol at 4 °C were stained by cytokeratin immunocytochemistry. CTCs were then detected and counted under microscopy by a pathologist, and then DNA was extracted from the CTCs on a glass slide (Figure 1b,c).

### 2.4. Immunocytochemistry Using Glass Slide CTC Specimens

Cytokeratin immunostaining of CTCs on the glass slide was carried out as follows: After treatment with a peroxidase-blocking (for HRP) and subsequent protein-blocking reagent, the CTC specimens were incubated with mouse monoclonal anti-pan-cytokeratin antibody (Oscar) for 1 h. After washing, specimens were incubated with an HRP (or AP)-labeled polymer-conjugated goat anti-mouse antibody (EnVision + system) (DAKO, CA, USA) for 30 min. After washing, color (brown or red) was developed using a Liquid DAB+substrate (or LPR, DAKO). Nuclei were counterstained by Meyer’s hematoxylin. The CTC glass slide was observed under a light microscope (Olympus BX50, Tokyo, Japan) to count cytokeratin-positive CTC numbers.

### 2.5. Immunocytochemistry-Based DNA Extraction from CTCs on a Glass Slide

Since DNA is known to be damaged by hydrogen peroxide and HRP systems [17,18], in this study, we used AP as the 2nd antibody-labeling system instead of HRP for DNA extraction. CTCs on the glass slide were immunostained by an anti-cytokeratin antibody, followed by incubation with AP-labeled polymer-conjugated goat anti-mouse antibody (EnVision+ system). After washing, the color (red) was developed using an LPR chromogen system for AP (DAKO). CTC number was counted and photographed under a light microscope. Then, DNA was extracted from the CTC circle (1 cm in diameter) on the glass slide using the QIAamp DNA micro kit (Qiagen, Valencia, CA, USA). EGFR mutation analysis, including Exon 19 deletion and Exon 21 mutations such as L856R, was conducted using a droplet digital PCR system (QX200, Bio-Rad Lab. Inc, Hercules, CA, USA). Surgically resected lung tissue specimens were analyzed for EGFR mutation by PCR assay (Cobas^®^ EGFR Mutation Test v2, Roche Diagnostics, Switzerland).

### 2.6. Statistical Analysis

The significance of differences between groups was determined by Student’s *t*-test and Welch’s test. A *p*-value of less than 0.05 was considered significant.

## 3. Results

### 3.1. New Platform for CTC Enumeration and Subsequent EGFR Mutation Analysis

After collection with filtration and subsequent transferring of CTCs to a glass slide (Figure 1a), fixed CTCs on glass slides were stained by cytokeratin immunocytochemistry using a secondary antibody labeled by either HRP or AP, followed by color (brown or red) development, respectively. No significant difference was observed between these two types of staining for CTC counting. Therefore, for the dual use of both enumeration and DNA extraction of CTCs, the method using AP-labeled secondary antibodies is preferable (Figure 1b,c).

The enumeration of CTCs on a glass slide under light microscopy appears to be more convenient and accurate than fluorescence staining of CTCs under a dark field. In our bright-field system, CTCs are easily identified as pan-cytokeratin-positive cells with atypical nuclear morphology (Figure 1b).

### 3.2. Patient Characteristics

Between 2021 and 2022, a total of 33 patients with non-small-cell lung cancer who underwent surgery at Teikyo University Hospital were enrolled in this study. Their clinical and pathological information is described in Table 1. Briefly, the average age of the patients was 72 years, and the male/female ratio was 18/15. The pathological stages of the patients with lung cancer were mostly Stage I, with two Stage II patients and one patient in Stage III. The histology of lung cancers included twenty-eight cases of adenocarcinoma, two cases of SCC, and three cases of other histologies, including pleomorphic carcinoma, metastatic colon carcinoma, and MALT lymphoma.

### 3.3. Enumeration of CTCs in PB, PA, and PV in Lung Cancer Patients

In these patients, 5 mL of blood was obtained from PB preoperatively, but from PV and PA, only 0.3–1.0 mL of blood could be collected immediately after the resection of the tumor-bearing lung. CTC numbers were measured using an immunocytology-based CTC detection platform under light microscopy, as described above. CTCs in PB were composed of a single cell with small cell clusters. In contrast, in PA and PV, most CTCs were detected as small and large cell clusters (Figure 2a). There was no significant difference between the number of CTCs in PA and PV (*p* = 0.9612), but the CTC numbers were significantly higher in PA than in PB (*p* < 0.01) and in PV than in PB (*p* < 0.05) (Figure 2b). In addition, it is noteworthy that numerous megakaryocytes were observed in PA, whereas few were observed in PV (Figure 2a).

### 3.4. Analysis of EGFR Mutations of CTCs in PV by Digital PCR

In this pilot study, we examined 12 blood samples from the PV of surgically resected lung tissue, because the CTC numbers were higher in PV compared to PB. CTCs collected from blood were stained by cytokeratin immunocytochemistry using AP-labeled secondary antibodies and subsequent permanent red (LPR) color development. After enumeration of CTCs, DNA was then quickly extracted and purified from CTCs located in the circle on the glass slide and then EGFR mutation was analyzed. Dots located in the left upper field, divided by the cross cursor, were considered positive signals in digital PCR (Figure 1c and Figure 2c).

The number of CTCs detected varied from 9 to 195 per blood sample across the 12 PV cases. Interestingly, the most abundant case (PV-4), with 195 CTCs, showed a significantly higher level of serum CEA compared to the other cases. The detected EGFR mutations included three Ex19del and three L858R mutations in 12 CTC samples and five Ex19del and three L858R mutations in 12 tumor tissue samples, respectively. Matched mutation-positive and -negative results in both CTCs and tissue were in 9 out of 12 cases. 

Two cases were unmatched and the remaining one case had controversial results with different EGFR mutations, probably due to the microheterogeneity of mutations (Table 2).

## 4. Discussion

We recently developed a unique immunocytology-based CTC detection platform that can simultaneously allow for both the enumeration and EGFR mutation analysis of CTCs under light microscopy. The key technology of our platform is the 3D metal filter used for the collection of CTCs and subsequent damage-free CTC transfer from the filter to the glass slide by an air pressure-mediated CTC transfer device [15,16]. To our knowledge, this CTC detection platform is the first device that utilizes the technical advantage of commercially available glass slides, which are critical for both the immunostaining and subsequent DNA extraction of CTCs. Furthermore, this is a convenient platform with high-cost performance, making it feasible for use in a clinical laboratory setting in hospitals, at least from a technical standpoint. In this study, using this platform, we found interesting results on the following two points. (1) Clear difference in CTC number/morphology among PB, PA, and PV. (2) Promising results of the EGFR mutation analysis of CTC using DNA extracted from CTCs on a glass slide.

In detail, we first found that the CTC number in PA and PV was almost comparable and was much higher than that in PB. Furthermore, CTCs in PV and PA are composed of mostly small and large clusters, whereas CTCs in PB are almost exclusively single cells with some small clusters. Such characteristics of CTCs in each type of blood is consistent with previous reports obtained from patients with breast cancer, colon cancer, and lung cancer, in whom CTCs in drainage vein blood were composed of small and large clusters, and their numbers were much higher than CTCs in PB [15,16,21,22]. Therefore, the low CTC number in PB is probably due to the dilution, destruction, and capillary trapping of CTCs during dynamic systemic circulation. In contrast, CTCs in the PA of primary lung tumors has not been examined, and this study is the first to trial this. Initially, we predicted that the CTC number in PA would be lower than that in PV. This is because PA has a thicker blood vessel wall than PV, making arterial blood more resistant to tumor invasion than PV. Unexpectedly, however, CTCs were also present abundantly in PA, at levels comparable to CTCs in PV of the lung. Recent reports by a respiratory surgeon suggest that early ligation of PV before PA can reduce the dissemination of tumor cells shed into peripheral blood during thoracoscopic surgery [23,24]. This idea is reasonable because CTCs are present abundantly in PA, like in PV, but they cannot enter peripheral blood due to the presence of the capillary vessel barrier. In addition, it is noteworthy that the lung is a specialized organ in which megakaryocytes derived from systemic bone marrow circulate through caval veins to PA and are finally trapped in the capillary vessels of the lung, as previously reported [19,20]. It is notable that in PA, megakaryocytes, which are atypical giant cells like tumor cells, are present abundantly in addition to CTCs. In contrast, in PV, CTCs are abundant, but the number of megakaryocytes is greatly reduced due to trapping by capillary vessels in the lung. This indicates a lower potential risk for diagnostic pitfalls of CTCs in PV compared to PA.

Second, before EGFR mutation analysis, we noted a previous report suggesting the possibility that DNA can be damaged during the usual immunostaining procedure in which H_2_O_2_ and HRP systems are involved in color development [17,18]. Based on these scientific reports, in this study, we used a hydroxyl radical-free AP-labeled polymer-conjugated secondary antibody system instead of HRP-labeled antibody/H_2_O_2_ for color development of CTCs. The result showed the secure acquisition of a sufficient amount of intact DNA and subsequent successful mutation analysis for 12 patients with lung cancer. This is probably due to the reduction in DNA damage during the immunostaining of a small number of CTCs by using an AP labeling system.

Mutation analysis of 12 lung cancer patients showed that EGFR mutations, including three Exon19del, three L858R, and six wild types, were detected in 12 CTC samples, whereas five Exon19del, three L858R, and four wild types were detected in the corresponding 12 tumor tissue samples. Nine out of twelve cases were matched, two cases were unmatched (wild type in CTC vs. mutated in tissue), and one case was controversial, with different EGFR mutations. In these two unmatched cases, the CTC numbers (18 and 34) were not so small. Therefore, the reason for these unmatched results is not simply due to a small CTC number but may be due to the DNA quality resulting from the bulk study in which DNA is extracted from CTCs and due to the varying number of contaminated leukocytes. The statistical data of the current mutation analysis are as follows: sensitivity: 0.75 (6/8), specificity: 1.0 (4/4), positive predictive value: 1.00 (6/6), and negative predictive value: 0.67 (4/6). These data indicate some promising results, but the sensitivity is still relatively low and there are two false-negative cases. In this respect, there is still room for improvement. The insufficient result of the present mutation analysis may be due to the low recovery rate and low quality of DNA extracted from CTCs with contaminated leukocytes. 

Recently, a highly sensitive method for single-cell RNA-sequence and DNA-sequence analysis of CTCs has become available [25,26,27,28]. Although there are some technical and cost-related issues, the integration of such a highly sensitive, single-cell method into our present CTC detection system may improve its sensitivity and allow for the mutation analysis of CTCs in PB in addition to PV. 

## 5. Conclusions

In conclusion, we developed a unique CTC detection platform that is useful for both the enumeration and genetic analysis of CTCs. Using this platform, sequential monitoring of CTC numbers in PB during drug therapy in lung cancer patients is now ongoing in our hospital as a clinical pilot study. However, as for genetic analysis based on PV, there are strong limitations, such as invasiveness for the patients and one-time analysis. Clinical studies of repetitive genetic analysis of CTCs from PB will take more time. A combination of the currently developing picking technology for single-cell CTCs or multiple pure CTCs under microscopy (but not bulk analysis like in the present study) and single-cell DNA sequence analysis would be a potentially useful tool for repetitive genetic analysis of CTCs in patients with lung cancer in the near future.

## Figures and Tables

**Figure 1 diagnostics-14-02064-f001:**
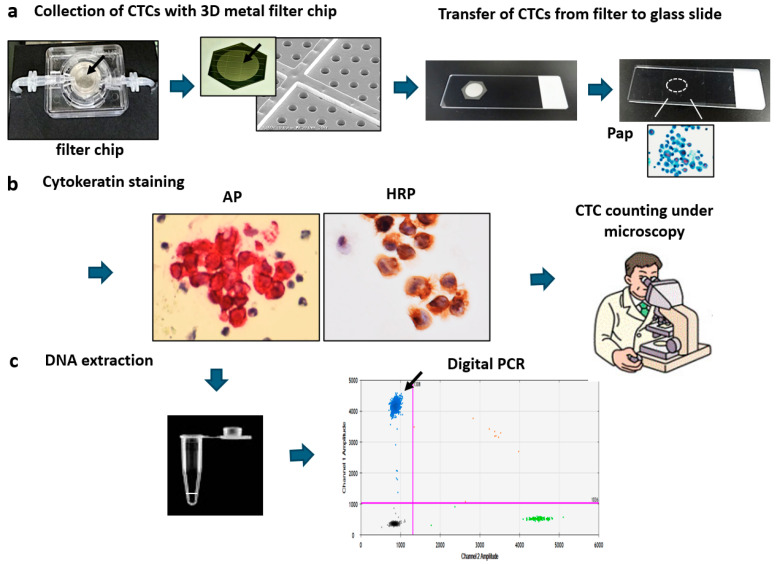
Overview of CTC enumeration and genetic analysis using an immunocytology-based CTC detection platform. (**a**) After the collection of CTCs on the 3D metal filter using an automated CTC enrichment apparatus, the filter is inverted onto a glass slide. The CTCs are then transferred and fixed to the glass slide using an air-pressure-mediated transfer apparatus. Finally, CTC slides are immediately stored in 95% ethanol at 4 ℃ until staining. (**b**) CTCs on a glass slide are stained by keratin immunocytochemistry using HRP- or AP-labeled secondary antibody until subsequent brown or red color development, respectively. Resultant keratin-positive CTCs with atypical nuclei are judged as CTCs and are counted under a light microscope. (**c**) DNA is extracted from CTCs on a glass slide, and EGFR mutation analysis is performed by digital PCR. Arrow indicates mutated droplets.

**Figure 2 diagnostics-14-02064-f002:**
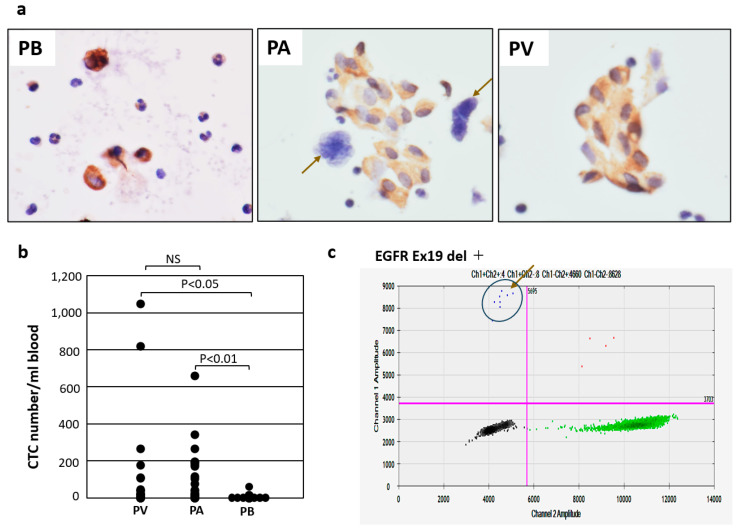
Characteristics of CTCs in PB, PA, and PV detected by cytokeratin immunocytochemistry and EGFR mutation analysis of CTCs in PV by digital PCR. (**a**) Morphology of CTCs stained by cytokeratin immunocytochemistry using HRP-labeled secondary antibodies and color development with DAB. CTCs in PB consist of single cells with some small clusters, whereas CTCs in PA and PV consist of large and small CTC clusters. It is of note that megakaryocytes are also seen in PA (arrows). (**b**) Quantitative comparison of CTC numbers in PA, PV, and PB. Statistically significant differences were observed between PA and PB (*p* < 0.01) and between PV and PB (*p* < 0.05). In contrast, no significant difference (NS) was observed between PA and PV (*p* = 0.9612). (**c**) Detection of EGFR mutation (Exon19del) by digital PCR using DNA extracted from CTCs in PV. Arrow indicates mutated droplets.

**Table 1 diagnostics-14-02064-t001:** Clinical characteristics of lung cancer patients included in this study.

Parameters		Number (*n* = 33)
Age (years, median, range)		72 (52–86)
Sex	Male	18 (54.5%)
	Female	15 (45.5%)
Location	Right	21 (63.6%)
	Left	12 (36.4%)
Invasion size (mm, median, range)		20 (2–48)
Lymph node Metastasis	pN1	2 (6.1%)
	pN2	1 (3.0%)
pStage	pStage I	28 (84.8%)
	pStage II	2 (6.1%)
	pStage III	1 (3.0%)
Histology	Adenocarcinoma	28 (84.8%)
	Squamous cell carcinoma	2 (6.1%)
	Others *	3 (9.1%)

* Others include pleomorphic carcinoma, metastatic colon carcinoma, and MALT lymphoma.

**Table 2 diagnostics-14-02064-t002:** Summary of EGFR mutations of CTCs in PV detected by digital PCR in 12 patients with lung cancer.

Case No	Serum CEA	CTC No	CTC-EGFR	Tissue-EGFR	Comparison
PV-1	2.7	12	L858R	Ex19del	Controversial
PV-2	3.7	18	wild type	Ex19del	Unmatched
PV-3	2.0	9	Ex19del	Ex19del	Correspond
PV-4	46.7	195	wild type	wild type	Correspond
PV-5	7.2	11	wild type	wild type	Correspond
PV-6	1.7	20	L858R	L858R	Correspond
PV-7	5.5	10	wild type	wild type	Correspond
PV-8	3.0	12	wild type	wild type	Correspond
PV-9	3.0	27	Ex19del	Ex19del	Correspond
PV-10	5.6	74	Ex19del	Ex19del	Correspond
PV-11	1.7	34	wild type	L858R	Unmatched
PV-12	3.6	85	L858R	L858R	Correspond

## Data Availability

The original contributions presented in the study are included in the article, further inquiries can be directed to the corresponding author.

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
