# Peer review of "Detection of Circulating Tumor Cells and EGFR Mutation in Pulmonary Vein and Arterial Blood of Lung Cancer Patients Using a Newly Developed Immunocytology-Based Platform"

_diagnostics, 2024, doi:10.3390/diagnostics14182064_

Round 1

Reviewer 1 Report

Comments and Suggestions for Authors

The article entitled, “Detection of circulating tumor cells and EGFR mutation in pulmonary vein and arterial blood of lung cancer patients using a newly developed immunocytology-based platform” by Dejima et al., presents a study on the detection of circulating tumor cells (CTCs) and EGFR mutations in lung cancer patients using a newly developed immunocytology-based platform. The importance of the present study is high because of the increasing importance of liquid biopsies in cancer diagnosis and management. The manuscript is generally well-written and organized but there are few areas that that require further clarification and inclusion of additional data. To improve the clarity and impact of the manuscript, the authors need to address the following comments:

1.     Few sentences in abstract section are slightly redundant, the authors should rephrase the same.

2.     The introduction provides a good background but lacks a clear hypothesis or research question. It would be beneficial to explicitly state the study's primary objectives.

3.     The authors should include more recent references when discussing the current limitations in CTC detection and EGFR mutation analysis.

4.     In the methods section, the authors should include clearer subheadings to provide the structured description of different procedures.

5.     A more detailed explanation of the statistical analysis and its appropriateness for the study would be beneficial.

6.     Though the results are presented clearly, but there is a lack of detailed statistical analysis in some sections. For example, the comparison of CTC numbers between different blood sources could be enhanced with more comprehensive statistical data.

7.     Although authors have interpreted the results obtained from the current study, the more critical analysis of the study’s limitations could be beneficial.

8.     Though the authors have discussion of the clinical implications of the findings, but it would be strengthened if more specific examples of how this platform could be integrated into current clinical practice.

The article can be accepted for publication after addressing the above-mentioned minor comments. 

Comments on the Quality of English Language

The manuscript is mostly well-written, but there are few areas where the language could be refined, such as:

On line 17: “CTC level were significantly higher” should be changed to “CTC levels were significantly higher”

There are few other places, where the language can be refined for better clarity. 

Author Response

Answer to reviewer comment 1

1. Few sentences in abstract section are slightly redundant, the authors should rephrase the same
Answer: According to the reviewer’s suggestion, we changed as follows in red.

Epidermal growth factor receptor (EGFR) tyrosine kinase inhibitors are powerful molecular targeting therapeutic agents for lung cancer. We recently developed an original immunocytology and glass slide-based circulating tumor cell (CTC) detection platform for both CTC enumeration and EGFR mutations analysis with DNA extracted from CTC. Using this platform, we conducted a pilot clinical study for CTC enumeration in peripheral blood (PB), pulmonary arterial blood (PA), and pulmonary venous blood (PV) from 33 patients with lung cancer (Stage I–III) who underwent surgery, followed by digital PCR-based EGFR mutation analysis of CTCs in PV from 12 patients. Results showed that CTC level were significantly higher in PV and PA than in PB (p<0.05, p<0.01, respectively), with a notable greater number of small and large CTC clusters. Genetic analysis of EGFR mutations of CTCs from PV (n=12) revealed 6 mutations including 3 Exon19del and 3 L856R in CTC and 8 EGFR mutations including 5 Exon19del and 3 L856R in lung tumor tissue. CTCs mutation status matched that in tissue samples in 9 patients, unmatched in 2 patients, and controversial in 1 patient, indicating a sensitivity of 0.75 (6/8) and specificity of 1.00 (4/4) with 2 false-negative result for CTC mutation analysis. This immunocytology-based CTC detection platform is a convenient method for detecting both CTC number and EGFR mutation status under microscopy, suggesting its potential as a liquid biopsy tool in the hospital for patients with lung cancer in some clinical settings.

2. The introduction provides a good background but lacks a clear hypothesis or research question. It would be beneficial to explicitly state the study's primary objectives

Answer: We agreed with the reviewer’s suggestion and added primary objectives of this study in the latter half of the introduction section as follows in red.

     Concerning the detection of CTC, numerous technologies have been developed. Most CTC detection platforms depend on multicolor immunofluorescence staining such as Keratin+/EpCAM+/CD45-/DAPI+ [11,12]. In contrast, reports on the immunocytology-based CTC detection method have been quite limited [13]. Recently, we developed a unique and convenient platform based on immunocytological detection of CTC on a glass slide under light microscopy [14,15,16]. In the present study, we newly developed a DNA extraction method after cytokeratin immunostaining of CTC on a glass slide through an alkaline phosphatase (AP) and LPR-based color development system. This is because AP does not induce DNA damage unlike the hydrogen-peroxide (H2O2)/horseradish peroxidase (HRP) system [17,18]. Primary objective of this study is to estimate the possibility whether our current multifunctional CTC detection system can be applicable to a new laboratory test for both enumeration and genetic analysis of CTC as an in-hospital CTC examination. For this purpose, using updated CTC detection system, we conducted a pilot study to detect EGFR mutations in 12 patients with lung cancer using digital PCR. In addition, we compared characteristics of CTCs in peripheral blood (PB), pulmonary arterial blood (PA), and pulmonary venous blood (PV) from 33 patients with Stage I–III lung cancer who underwent surgery. We previously reported the presence of abundant CTCs in venous drainage blood of patients with breast and colon cancer [15,16], as well as the presence of megakaryocytes in arterial and venous blood of patients with lung cancer [19,20]. The hemodynamic aspect of CTCs in the pulmonary circulation including PA, PV and PB and the potential diagnostic pitfall of CTCs by megakaryocytes in PA and PB are discussed.

3. The authors should include more recent references when discussing the current limitations in CTC detection and EGFR mutation analysis

Answer: For discussion the current possibility and limitation of liquid biopsy, we added 2 recent review articles (2023-24) in the introduction section and 1 recent original article (2022) for single cell mutation analysis of CTC in the Discussion section.

1) Alexander Ring, Bich Doan Nguyen-Sträuli, Andreas Wicki, Nicola Aceto. Biology, vulnerabilities and clinical applications of circulating tumor cells. Nature Reviews Cancer., 2023, 23: 95 –111.

2) Isabel Heidrich, Carmen M.T. Roeper, Charlotte Rautmann, Klaus Pantel, Daniel J. Smit. Liquid Biopsy–A new diagnostic concept in oncology. Laryngorhinootologie, 2024, 103(01): 40-46.

3) Negishi, R.; Yamakawa, H.; Kobayashi, T.; Horikawa, M.; Shimoyama, T.; Koizumi, F.; Sawada , T.; Oboki, K.; Omuro, Y.; Yoshino, T.; et al. Transcriptomic profiling of single circulating tumor cells provides insight into human metastatic gastric cancer. Communications Biology., 2022. 5: 20. https://doi.org/10.1038/s42003-021-02937-x

4. In the methods section, the authors should include clearer subheadings to provide the structured description of different procedures

Answer: According to the reviewer’s comment, we added subheading and concise explanation of each procedures with several original devices as following three steps. 1) Filtration-based CTC collection system. This is composed of microfluidic chip (filter chip), containing a nickel filter with 8 μm pores and 3-dimensional (3D) lattice structure of 10 nm thickness (Optnics Precision Co., Ltd., Tochigi, Japan) (Fig. 1a). Four filter chips can be run simultaneously by the automated CTC enrichment apparatus with a fluid pressure control system [15] (Maruyasu Industries Co., Ltd., Okazaki, Japan). After applying whole blood to this apparatus, collected CTCs on the filter are washed with PBS/EDTA solution, and subsequent fixation of CTCs with 10 % formalin solution for 20 min. 2) CTC transfer system to the glass slides. The 3D metal filter was detached from the filter chips and was placed upside down on a coated glass slide (MAS coat, Matsunami, Osaka, Japan) [17] and transferred to a coated glass slide by the air pressure-mediated transfer apparatus (data not shown) (Maruyasu Industry Co., Ltd). 3) Immunocytochemistry of CTCs on a glass slide. CTC glass slide stored in 95% ethanol at 4℃ were stained by cytokeratin Immunocytochemistry. CTCs were then detected and counted under microscopy by pathologist and then DNA was extracted from CTCs on a glass slide. (Fig. 1b and 1c).

5. A more detailed explanation of the statistical analysis and description of different appropriateness for the study would be beneficial.

Answer: According to the reviewer’s suggestion, we explained the statistical analysis with more detail and discussed the possible reason for this immature results.

Statistical analysis of current EGFR mutation test (n=12) is as follows; sensitivity: 0.75 (6/8), specificity: 1.0 (4/4), positive predictive value: 1.00 (6/6) and negative predictive value: 0.67 (4/6), indicating some promising results, but sensitivity is still relatively low (0.75) and there are 2 false-negative cases. In this respect, there is still room for improvement. Insufficient result of the present mutation analysis by digital PCR may be due to the low recovery rate and low quality of DNA extracted from CTCs with contaminated leukocytes.

6. Though the results are presented clearly, but there is a lack of detailed statistical analysis in some sections. For example, the comparison of CTC numbers between different blood sources could be enhanced with more comprehensive statistical data.

Answer: In Figure 2b, some typographical errors are found, and we changed Figures 2b according to the detailed analysis of CTC number between PA vs PV, PA vs PB and PV vs PB. In the Figure 2 legend, we added real p values as follows; Statistically significant differences were observed between PV vs PB (p<0.05) and PA vs PB (p<0.01). In contrast, no significant difference (NS) was observed between PA vs PV (p=0.9612).

7. Although authors have interpreted the results obtained from the current study, the more critical analysis of the study’s limitations could be beneficial.

Answer: Most critical limitation of our present CTC detection system is one-time analysis using PV with increased CTC number, but not repetitive analysis like PB. This limitation of our system can be improved by the single cell DNA-sequence analysis. This limitation with our CTC detection system is described in the 5 Conclusion section.

8. Though the authors have discussion of the clinical implications of the findings, but it would be strengthened if more specific examples of how this platform could be integrated into current clinical practice.

Answer: Sequential monitoring of CTC number in PB during drug therapy in lung cancer patients is now on going in our hospital as a clinical pilot study. However, as for repetitive genetic analysis of CTC using PB, it will take a more time for clinical study. Combination of now developing technology for picking single cell CTC or pure CTC clusters under microscopy (but not bulk analysis) and single cell DNA sequence analysis would be a potentially useful tool for repetitive genetic analysis with CTC in patients with lung cancer in the near future. We added these discussion in the Conclusion section.

Reviewer 2 Report

Comments and Suggestions for Authors

I read with interest the article entitled “Detection of circulating tumor cells and EGFR mutation in pul-2 monary vein and arterial blood of lung cancer patients using a 3 newly-developed immunocytology-based platform”. In this manuscript, Dejima and colleagues report results from a pilot study aiming to the isolation and enumeration of circulating tumor cells (CTCs) from different body anatomical districts (i.e. peripheral blood PB, pulmonary arterial blood PA, and pulmonary venous blood PV), also for EGFR detection in patients with stage I-III stage lung cancer. The authors found that the rate of CTC-positive patients was higher when considering enumeration from PV and PA blood with respect to PB, while mutational analysis of the EGFR gene revealed a concordance with tumor tissue in 9 out of 12 cases, with 75% sensitivity and 100% specificity. Although limited, the number of enrolled patients is appropriate for a pilot study. The study is of interest to the field, particularly in comparing different CTC sources, given the clinical context. However, this reviewer has some concerns to be addressed by the authors.

·  The authors state that the rate of CTC-positive patients was higher when considering PA and PV compared to PB. This is an intriguing finding, especially given that the rate of positivity in stage I-III lung cancer patients is generally low. Did they compare PB, PV, and PA from the same patients? If so, they should provide a more detailed description of the data regarding these enumerations.

·  In the Methods section, the authors report that they used the QIAamp DNA Micro Kit for DNA extraction. According to the kit protocol, the procedure is suitable for extraction from "small volumes of blood, blood cards, urine, small tissue samples, including laser microdissections." Could the authors be more specific about the protocol they followed? Additionally, considering that a column-based method is not always ideal for a small number of cells, what was the minimum number of cells from which they obtained sufficient evaluable DNA?

·          As the authors state, liquid biopsy is critical for clinical practice in NSCLC patients. This is important especially in advanced stages (10.21037/tlcr.2019.04.14). While for EGFR detection ctDNA is a well recognized biological material for molecular testing, CTCs could have a role in detecting other tumor oncogene addictions, i.e. gene fusions, considering that circulating RNA is not feasible to this aim (doi 10.21037/tlcr.2019.09.15).

·  In Table 1, the authors should also report the percentage of patients for each clinical parameter.

·  In Table 2, "wild" should be corrected to "wild type."

·  One advantage of measuring the number of CTCs in peripheral blood is that the procedure is minimally invasive and feasible for most patients, allowing for serial disease monitoring. Conversely, while the CTC count was higher in PA and PV, these sources are not minimally invasive. The authors should explain the advantages of counting CTCs in PA and PV from a translational perspective.

·  In the Introduction, the authors discuss liquid biopsy as involving cell-free DNA (cfDNA), circulating tumor DNA (ctDNA), and CTCs, but exosomes and extracellular vesicles are also relevant in liquid biopsy. I suggest to add the usefulness of these biological vesicles for translational research. (doi: 10.3390/cells13040337).

Comments on the Quality of English Language

There are some typing errors to be addressed

Author Response

 Answer to reviewer comment 2

1. The authors state that the rate of CTC-positive patients was higher when considering PA and PV compared to PB. This is an intriguing finding, especially given that the rate of positivity in stage I-III lung cancer patients is generally low. Did they compare PB, PV, and PA from the same patients? If so, they should provide a more detailed description of the data regarding these enumerations.

Answer: In this study, we compared PB, PV, and PA from the same patients. Higher CTC number of drainage vein blood from the primary tumour than peripheral blood is well known event reported previously in colorectal cancer and breast cancer. According to the reviewer’s question we, added these sentences in the Result section and Discussion section as follows;

2. In the Methods section, the authors report that they used the QIAamp DNA Micro Kit for DNA extraction. According to the kit protocol, the procedure is suitable for extraction from "small volumes of blood, blood cards, urine, small tissue samples, including laser microdissections." Could the authors be more specific about the protocol they followed? Additionally, considering that a column-based method is not always ideal for a small number of cells, what was the minimum number of cells from which they obtained sufficient evaluable DNA?

Answer: We used this kit according to the procedure for laser microdissections. We drop proteinase-K soln on the CTCs cluster fixed on a glass slide and dissolve CTCs well and recover to PCR tube. In a column-based method, the minimum number of CTCs from which they obtained sufficient DNA is approximately 10 cells. We added these detail partly in the Result section.

3. As the authors state, liquid biopsy is critical for clinical practice in NSCLC patients. This is important especially in advanced stages (10.21037/tlcr.2019.04.14). While for EGFR detection ctDNA is a well-recognized biological material for molecular testing, CTCs could have a role in detecting other tumor oncogene addictions, i.e. gene fusions, considering that circulating RNA is not feasible to this aim (doi 10.21037/tlcr.2019.09.15).

Answer: We agree with the reviewer’s idea. We added future potential application of single cell to oligo cells (CTCs)-RNA seq analysis and FISH analysis, as advantages of CTC for molecular diagnostics in the Discussion section. We also referred the following paper by

4. In Table 1, the authors should also report the percentage of patients for each clinical parameter. In Table 2, "wild" should be corrected to "wild type."

Answer: According to the reviewer's comment, we added and changed words in the Table 1, 2

5. One advantage of measuring the number of CTCs in peripheral blood is that the procedure is minimally invasive and feasible for most patients, allowing for serial disease monitoring. Conversely, while the CTC count was higher in PA and PV, these sources are not minimally invasive. The authors should explain the advantages of counting CTCs in PA and PV from a translational perspective.

Answer: We agree with the reviewer’s comment. Counting CTCs in PA and PV is not a repetitive method, therefore is not suitable for clinical use. In the present paper, we used CTC-rich blood (PA, PV) to certify the potential role of our new CTC detection platform for both enumeration and genetic analysis.

6. In the Introduction, the authors discuss liquid biopsy as involving cell-free DNA (cfDNA), circulating tumor DNA (ctDNA), and CTCs, but exosomes and extracellular vesicles are also relevant in liquid biopsy. I suggest to add the usefulness of these biological vesicles for translational research. (doi: 10.3390/cells13040337)

Answer: We agree with reviewer’s comment and add relevance of exosomes and extracellular vesicles in liquid biopsy in the introduction section with following Reference (5); Bandini , S.; Ulivi , P.; Rossi T. Extracellular Vesicles, Circulating Tumor Cells, and Immune Checkpoint Inhibitors: Hints and Promises. Cells. 2024, 13, 337.
